# Comparative analysis of protein expression between oesophageal adenocarcinoma and normal adjacent tissue

Ben Nicholas[1,2*☉], Alistair Bailey[1,2☉], Katy J. McCann[3], Robert C. Walker[3], Peter Johnson[4], Tim Elliott[2,5], Tim J. Underwood[3], Paul Skipp[1]

1 Centre for Proteomic Research, Biological Sciences and Institute for Life Sciences, Building 85, University of Southampton, Southampton, United Kingdom, 2 Centre for Cancer Immunology and Institute for Life Sciences, Faculty of Medicine, University of Southampton, Southampton, United Kingdom, 3 School of Cancer Sciences, Faculty of Medicine, University of Southampton, Southampton, United Kingdom, 4 Cancer Research UK Clinical Centre, University of Southampton, Southampton, United Kingdom, 5 Oxford Cancer Centre for Immuno-Oncology and CAMS-Oxford Institute, Nuffield Department of Medicine, University of Oxford, Southampton, United Kingdom

☉ These authors contributed equally to this work.
* bln1@soton.ac.uk

## Abstract

Oesophageal adenocarcinoma (OAC) is the 7th most common cancer in the United Kingdom (UK) and remains a significant health challenge. This study presents a proteomic analysis of seven OAC donors complementing our previous neoantigen identification study of their human leukocyte antigen (HLA) immunopeptidomes. Our small UK cohort were selected from donors undergoing treatment for OAC. We used label-free mass spectrometry proteomics to compare OAC tumour tissue to matched normal adjacent tissue (NAT) to quantify expression of 3552 proteins. We identified differential expression of a number of proteins previously linked to OAC and other cancers including common markers of tumourigenesis and immunohistological markers, as well as enrichment of processes and pathways relating to RNA processing and the immune system. Our findings also offer insight into the role of the protein stability in the generation of an OAC neoantigen we previously identified. These results provide independent corroboration of existing oesophageal adenocarcinoma biomarker studies that may inform future diagnostic and therapeutic research.

## Introduction

Oesophageal adenocarcinoma (OAC) accounts for about 2% of all cancer diagnosis in the UK, with an increase in 10-year survival from 4% to 12% in the last 50 years [1]. Treatment options centre on resection of the oesophagus in early-stage OAC, and chemo- or radio-therapy combined with surgery for later stage OAC [2]. Previously we presented proof-of-concept findings using mass spectrometry proteomics to identify human leukocyte antigen (HLA) presented neoantigens from a cohort of OAC donors as targets for cancer vaccines [3]. In the UK population, the most recent data for proportions of tumour stage at diagnosis were I = 5.8%, II = 11.4%, III = 24.4%, IV = 38% and 20.4% unknown. 69% of diagnosis

**Data availability statement:** The mass spectrometry proteomics data have been deposited to the ProteomeXchange Consortium via the PRIDE partner repository with the dataset identifier PXD054428 and 10.6019/PXD054428

**Funding:** This study was supported by a Cancer Research UK Centres Network Accelerator Award Grant (C328/A21998). Instrumentation in the Centre for Proteomic Research is supported by the Biotechnology and Biological Sciences Research Council (BM/M012387/1). The funders had no role in study design, data collection and analysis, decision to publish, or preparation of the manuscript.

**Competing interests:** The authors have declared that no competing interests exist.

were in men and 31% in women, with a median age at diagnosis, of 71 [4]. Here we present results of a small selected sample from the underlying OAC population of 7 OAC donors who were all undergoing surgery as part of their treatment. They comprised tumour stages II = 28.6%, III = 57.1% and IV = 14.3% and our samples were all from men with a lower median age at diagnosis of 68 (S1 Table). Rather than population level inferences, our focus here was on characterising tumour tissues. We performed a comparative proteomic analysis of OAC tumour tissue to matched normal adjacent tissue (NAT). Using label free quantification (LFQ) of bottom-up mass spectrometery proteomics we sought to identify differential expression of proteins (DEP) between OAC and NAT that may inform diagnosis and treatment options.

## Results

We quantified 3552 proteins across 7 patients using label free quantification (LFQ) [5,6] yielding protein identifications from the normalised top 3 peptide intensities (S2 Table). To confirm we could distinguish between OAC and NAT tissues using protein expression we performed Principal Component Analysis (PCA) using the normalised top 3 peptide intensities of the 500 most variable proteins (Fig 1) [7]. The PCA yielded clear separation between OAC and NAT along PC1 accounting for 45% of the variance between the tissues. However, whilst the NAT samples were tightly grouped, the tumour samples were more dispersed, indicating some heterogeneity between tumours, most notably donor EN-430-11 (Fig 1A). Plotting the PCA loadings to examine the expression of proteins driving the separation towards the OAC samples indicated three proteins: Keratin, type I cytoskeletal 18 (K1C18), Anterior gradient protein 2 homolog (AGR2) and Gamma-interferon-inducible lysosomal thiol reductase (GILT) (Fig 1B). Over 90% of the variation between the matched OAC and NAT samples was accounted for by the first 10 principal components (Fig 1C).

We then grouped the samples according to OAC or NAT and calculated differential protein expression (DEP) [8]. Of the 3552 proteins, we found 419 DEPs for OAC and 40 DEPs for NAT at thresholds of a $\log_2$ fold-change of greater than 1 and p-value of less than 1% (Fig 2A). These and the other thresholds used here are necessarily arbitrary and chosen to balance being conservative whilst not over-excluding information. The data without thresholds is provided in Supporting Information S3 Table.

As indicated by the PCA, K1C18 and AGR2 were the most DEP for OAC. We identified high expression in OAC of Endoplasmic reticulum chaperone BiP (HSPA5), Deoxynucleoside triphosphate triphosphohydrolase SAMHD1 (SAMHD1), Rho GDP-dissociation inhibitor 2 (ARHGDIB). Other notable OAC DEPs were Cell division cycle 5-like protein (CDC5L), Metalloproteinase inhibitor 1 (TIMP1), Matrix metalloproteinase-9 (MMP9) and Lysosome-associated membrane glycoprotein 1 (LAMP1) (S3 Table). Notable in NAT were high expression of Protein-glutamine gamma-glutamyltransferase E (TGM3) and Heat shock protein beta-1 (HSPB1).

Fig 2B focuses in on the most statistically significant DEPs (92 proteins below FDR of 2%) across the OAC cohort. We found high expression for proteins in OAC relating to cell structure such as Keratin, type II cytoskeletal 8 (K2C8), RNA processing and protein folding such as Nucleolar RNA helicase 2 and (DDX21), Peptidyl-prolyl cis-trans isomerase (FKB11), and the immune system such as antigen peptide transporters 1 & 2 (TAP1, TAP2), Mucin-1 (MUC1) and Intercellular adhesion molecule 1 (ICAM1). Collectively these are all proteins that impact cellular phenotype and immunoregulation. Donor EN-430-11, an outlier on the PCA plot, has relatively low expression of the proteins K1C18, AGR2 and GILT that drive the separation between the OAC and NAT PCA (Fig 1A).

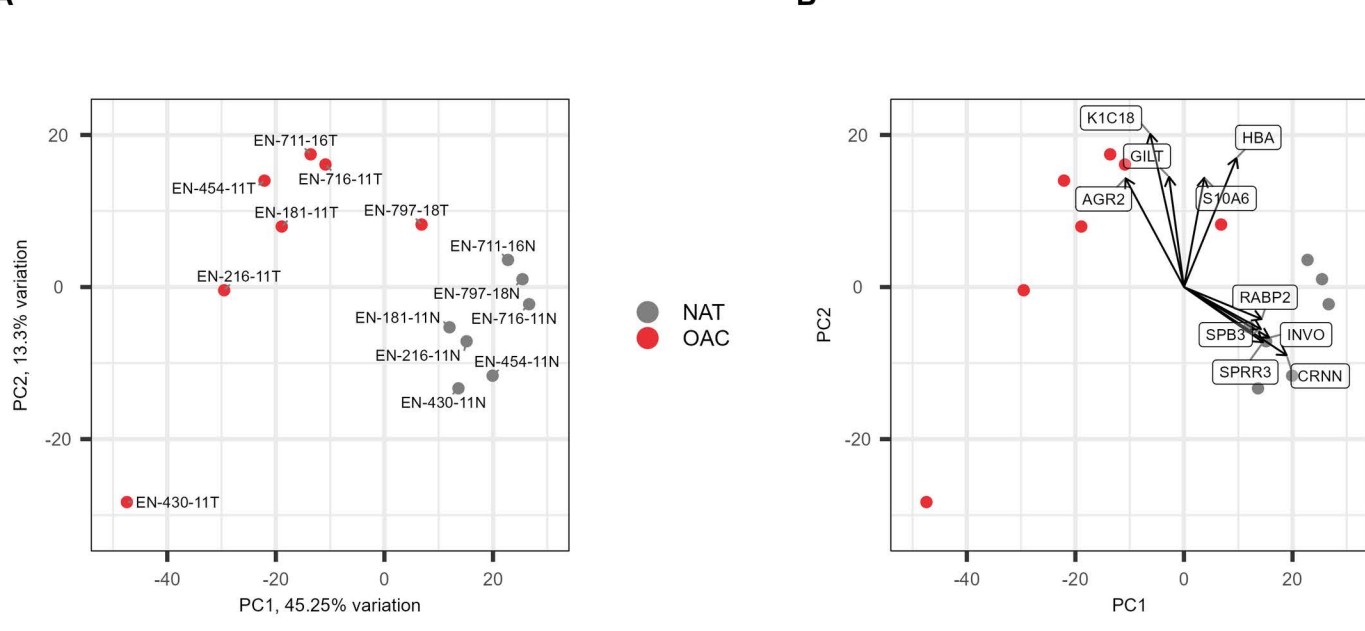

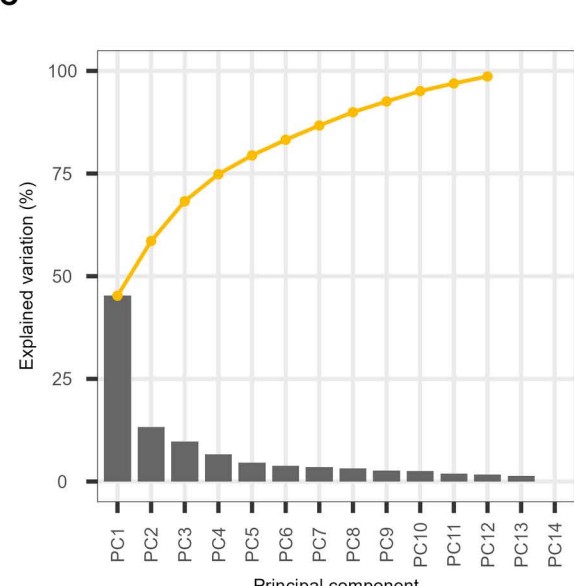

**Fig 1. Principal component analysis of OAC and NAT.** (A) PCA of normalised top 3 peptide intensities of 500 most variable proteins. OAC (red) & NAT (grey). Samples are numbered with donor identifier. (B) PCA plot with the loadings from proteins contributing to PC1 and PC2. (C) Scree plot of contribution of principal components to total variance.

A striking observation is for donor EN-454-11 and Nucleolar protein 58 (NOP58) which was generally highly expressed in OAC except for EN-454-11 (Fig 2B, S2 Table). We previously reported direct observation a putative neoantigen eluted from tumour HLA for EN-454-11 derived from mutation G95R in NOP58 [3]. Using DDGun [9] we calculated a change in the Gibbs free energy of unfolding ($\Delta\Delta G$) between the wild type and G95R mutant NOP58

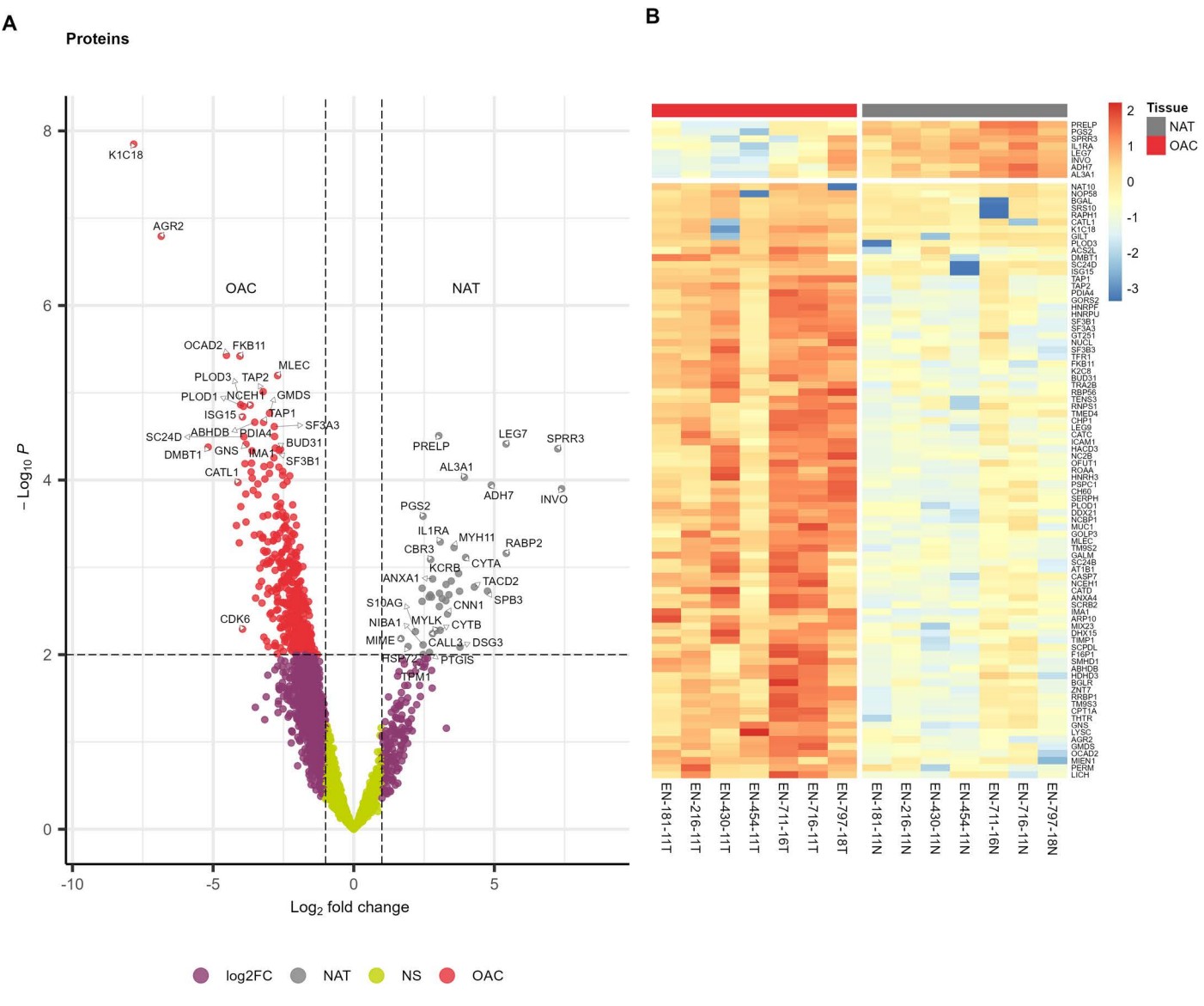

**Fig 2. Differential protein expression of OAC and NAT.** (A) Volcano plot of differentially expressed proteins for OAC vs NAT. Proteins are labelled with gene names. Thresholds are represented by dotted lines at p-value of 1% and log2 fold change of 1. (B) Heatmap of DEPs below a FDR of 2% (n = 92). Colour bar shows $\log_2$ fold change rescaled as z-scores, i.e., each unit from zero represents one standard deviation from the row average value for each protein.

protein of −0.5 kcal/mol, indicating a decrease in the stability of NOP58 expressed by donor EN-454-11 (Supporting Information).

Finally we performed functional analysis using 232 OAC DEPs ($\log_2$ fold-change greater than 2 and below FDR 5%). We identified greatest enrichment for pathways of biological processes relating to RNA processing, particularly mRNA splicing (Fig 3A, S4 Table). For enrichment of Reactome pathways [10] we also found changes in RNA processing, but also in immunological pathways, specifically neutrophil degranulation and antigen processing and presentation (Fig 3B, S4 Table).

**A**

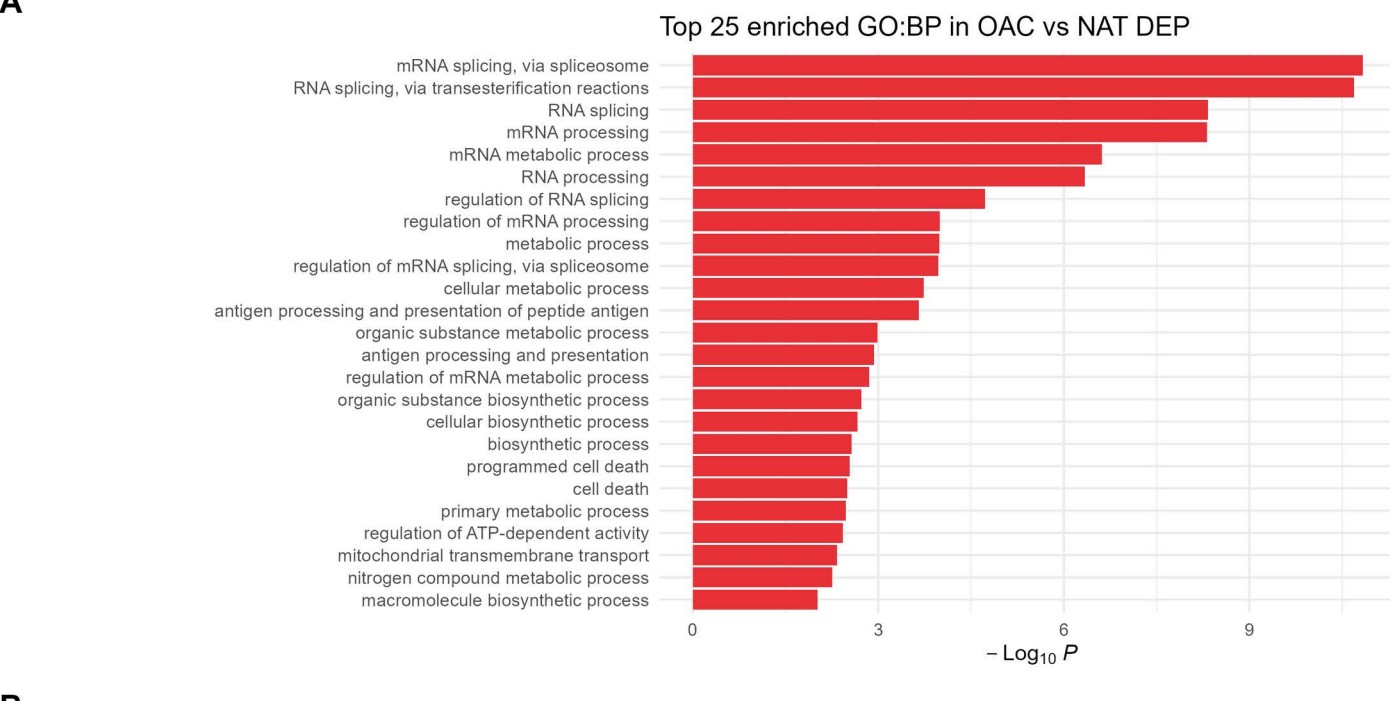

**B**

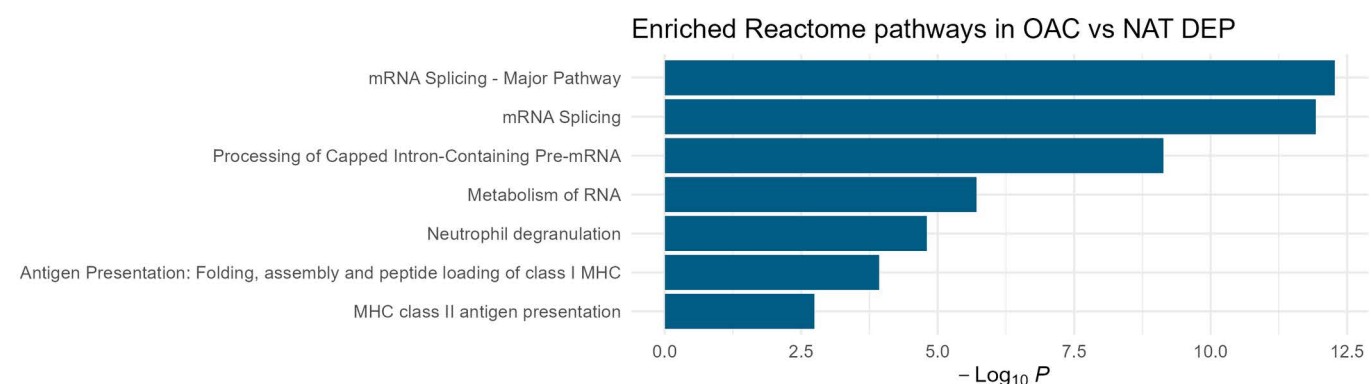

**Fig 3. Functional analysis of OAC DEPs.** (A) Enriched GO Biological Processes. The top 25 pathways are shown. (B) Enriched Reactome pathways. Statistical significance level indicated by the -$\log_{10}$ p-value on the x-axis. Proteins were selected using thresholds for OAC DEPs above a $\log_2$ fold change 2 and below a FDR of 5% (n = 232).

## Discussion

Previously we used proteogenomics analysis to identify patient specific neoantigens arising OAC mutations as therapeutic targets [3]. Here we compared the proteomes of OAC tissues to NAT from seven patients of the same cohort. This was a small study, limiting the extent to which our findings can be generalised. For example, epithelial cell adhesion molecule (EPCAM) was not uniformly expressed across all samples, so we are unable to confirm reports of high expression of EPCAM as a putative OAC biomarker [11,12]. The other main limitation in our design is that we are unable to compare these differential protein expression observations to other information that might support or discount their value, such as gene expression or comparison to oesophageal squamous cell carcinoma tissue.

The principal value of the results presented here is the quantification of 3,500 proteins of which nearly 500 were differently expressed as a resource to other OAC researchers. Amongst the OAC DEPs we found high expression of common markers of tumourigenesis, including AGR2 and keratin K1C18 [13] and that along with antigen processing related protein GILT, these proteins drove the separation between OAC and NAT in the Principal Component Analysis. AGR2 is a known unfavourable prognostic marker in renal and liver cancer and OAC [11,13,14]. Moreover, the role of AGR2 in tumourigenesis in the oesophagus has previously been seen in higher gene expression of AGR2 in the OAC precursor condition Barrett's oesophagus with respect to NAT [15], and increased expression of AGR2 in fibroblast cells was seen to promote tumour growth in mice [16]. We also observed differential expression in OAC of putative immunohistological markers BiP (HSPA5), SAMHD1 (SAMHD1) and Rho GDP-dissociation inhibitor 2 (ARHGDIB) [11]. Other DEPs of interest included G2/M checkpoint related protein Cell division cycle 5-like protein (CDC5L), a putative target for checkpoint inhibition [17], and putative prognostic biomarkers in colorectal, breast and ovarian cancers: Metalloproteinase inhibitor 1 (TIMP1), Matrix metalloproteinase-9 (MMP9) [18] and Lysosome-associated membrane glycoprotein 1 (LAMP1) [19]. We identified differential expression in NAT of Protein-glutamine gamma-glutamyltransferase E (TGM3) and Heat shock protein beta-1 (HSPB1) consistent with previous reports [11,20].

Additionally, the finding that the G95R mutation decreases the stability of NOP58 suggests that for donor EN-454-11, NOP58 is more likely to be a defective ribosomal product (DRiP) [21,22]. This on the one hand may explain the decreased expression seen in Fig 2B for donor EN-454-11, whilst on the other hand have increased the probability of NOP58 neoantigen presentation that we previously observed [3]. It suggests utility in quantifying the affects of single nucleotide polymorphisms on protein stability and subsequent processing in identifying putative DRiP-derived neoantigens.

Our functional analysis corresponds with previous reports characterising changes in OAC tissues in identifying enriched biological processes and Reactome pathways for mRNA processing and antigen processing [17]. Additionally Reactome pathway enrichment of the neutrophil degranulation pathway is indicative of inflammation, a known risk factor in cancer [23].

Overall, these observations offer independent corroboration and contrast to existing studies seeking to identify biomarkers or targets for more effective OAC specific treatments, and a catalogue of additional putative biomarkers of OAC which can be validated in larger cohorts in the future. Moreover our NOP58 observation suggests another parameter, protein stability, that can be used in the prediction of putative neoantigens for personalised therapies.

## Materials and Methods

### Ethics statement

Informed written consent was provided for participation by all individuals. Ethical approval for this study was granted by the Proportionate Review Sub-Committee of the North East – Newcastle & North Tyneside 1 Research Ethics Committee (Reference 18/NE/0234). This study was approved by the University of Southampton Research Ethics Committee. For the study presented here samples were accessed on 2nd October 2018, and only authors RCW and TJU had access to information that could identify individual participants during or after data collection.

### Tissue preparation

Subjects diagnosed with OAC were recruited to the study (see S1 Table for clinical characteristics). Tumours were excised from resected oesophageal tissue post-operatively by pathologists

and processed either for histological evaluation of tumour type and stage, or snap frozen at −80°C.

## Protein extraction and digestion

Snap frozen tissue samples were briefly thawed and weighed prior to 30s of mechanical homogenization (using disposable probes, Fisher, UK) in 4mL lysis buffer (0.02M Tris, 0.5% [w/v] IGEPAL, 0.25% [w/v] sodium deoxycholate, 0.15 mM NaCl, 1mM ethylenedi-aminetetraacetic acid (EDTA), 0.2 mM iodoacetamide supplemented with EDTA-free prote-ase inhibitor mix). Homogenates were clarified for 10 min at 2000$g$, 4°C and then for a further 60 min at 13,500$g$, 4°C.

Protein concentration of tissue lysates was determined by BCA assay, and volumes equivalent to 100 mg of protein were precipitated using methanol/chloroform as previously described [24]. Pellets were briefly air-dried prior to resuspension in 6 M urea/50 mM Tris-HCl (pH 8.0). Proteins were reduced by the addition of 5 mM (final concentration) DTT and incubated at 37°C for 30 min, then alkylated by the addition of 15 mM (final concentration) iodoacetamide and incubated in the dark at room temperature for 30 min. 4 µg Trypsin/LysC mix (Promega, UK) were added and the sample incubated for 4 h at 37°C, then 6 volumes of 50 mM Tris-HCl pH 8.0 were added to dilute the urea to < 1 M, and the sample was incubated for a further 16 h at 37°C. Digestion was terminated by the addition of 4 µL of TFA, and the sample clarified at 13,000×$g$ for 10 min at RT. The supernatant was collected and applied to Oasis Prime microelution HLB 96-well plates (Waters, UK) which had been pre-equilibrated with acetonitrile. Peptides were eluted with 50 µL of 70% acetonitrile and dried by vacuum centrifugation prior to resuspension in 0.1% formic acid.

## Mass spectrometry proteomics

8 µg of peptides per sample were separated by an Ultimate 3000 RSLC nano system (Thermo Scientific, UK) using a PepMap C18 EASY-Spray LC column, 2 µm particle size, 75 µm × 75 cm column (Thermo Scientific, UK) in buffer A (H$_2$O/0.1% Formic acid) and coupled on-line to an Orbitrap Fusion Tribrid Mass Spectrometer (Thermo Fisher Scientific, UK) with a nano-electrospray ion source.

Peptides were eluted with a linear gradient of 3–30% buffer B (acetonitrile/0.1% formic acid) at a flow rate of 300 µL/min over 200 min. Full scans were acquired in the Orbitrap analyser in the scan range 300–1,500 m/z using the top speed data dependent mode, perform-ing an MS scan every 3 second cycle, followed by higher energy collision-induced dissociation (HCD) MS/MS scans. MS spectra were acquired at a resolution of 120,000, RF lens 60% and an automatic gain control (AGC) ion target value of 4.0e5 for a maximum of 100 ms. MS/MS scans were performed in the ion trap, higher energy collisional dissociation (HCD) fragmen-tation was induced at an energy setting of 32% and an AGC ion target value of 5.0e3.

## Proteomic data analysis

Raw spectrum files were analysed using Peaks Studio 10.0 build 20190129 [5,25] and the data processed to generate reduced charge state and deisotoped precursor and associated product ion peak lists which were searched against the UniProt database (20,350 entries, 2020-04-07) plus the corresponding mutanome for each sample (~1,000–5,000 sequences) and con-taminants list in unspecific digest mode. Parent mass error tolerance was set a 10 ppm and fragment mass error tolerance at 0.6 Da. Variable modifications were set for N-term acetyl-ation (42.01 Da), methionine oxidation (15.99 Da), carboxyamidomethylation (57.02 Da) of cysteine. A maximum of three variable modifications per peptide was set. The false discovery

rate (FDR) was estimated with decoy-fusion database searches [5] and were filtered to 1% FDR. Data was deposited in PRIDE [26].

### Differential protein expression

Label free quantification using the Peaks Q module of Peaks Studio [5,6] yielding matrices of protein identifications as quantified by their normalised top 3 peptide intensities. The resulting matrices were filtered to remove any proteins for which there were more than two missing values across the samples. Differential protein expression was then calculated with DEqMS using the default parameters [8].

Principal component analysis of the normalised top 3 peptide intensities was performed using DESEq2 [7] and PCATools [27].

Results were visualised using EnhancedVolcano [28], pheatmap [29] and ggplot2 [30].

### Functional analysis

Functional enrichment analysis was performed using g:Profiler [31] using default settings for homo sapiens modified to exclude GO electronic annotations. Protein ids were used as inputs.

## Supporting Information

**S1 Table. Patient information for the 7 male donors in this study**
(CSV)

**S2 Table. Peaks normalised top 3 peptide intensities.**
(CSV)

**S3 Table. Output of DEqMS.**
(CSV)

**S4 Table. g:Profiler outputs.**
(CSV)

## Author contributions

**Conceptualization:** Ben Nicholas, Alistair Bailey, Paul Skipp.

**Data curation:** Alistair Bailey.

**Formal analysis:** Ben Nicholas, Alistair Bailey.

**Funding acquisition:** Peter Johnson, Tim Elliott, Paul Skipp.

**Investigation:** Ben Nicholas, Alistair Bailey, Katy J McCann, Robert C. Walker, Tim J. Underwood.

**Methodology:** Ben Nicholas, Alistair Bailey, Paul Skipp.

**Project administration:** Alistair Bailey, Katy J McCann, Peter Johnson, Tim Elliott, Paul Skipp.

**Resources:** Katy J McCann, Robert C. Walker, Tim J. Underwood, Paul Skipp.

**Supervision:** Peter Johnson, Tim Elliott, Tim J. Underwood, Paul Skipp.

**Visualization:** Alistair Bailey.

**Writing – original draft:** Ben Nicholas, Alistair Bailey.

**Writing – review & editing:** Ben Nicholas, Alistair Bailey.

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
