## [Decision Letter · Decision Letter 0]

4 Nov 2024

Dear Dr. Bailey,

Thank you for submitting your manuscript to PLOS ONE. After careful consideration, we feel that it has merit but does not fully meet PLOS ONE’s publication criteria as it currently stands. Therefore, we invite you to submit a revised version of the manuscript that addresses the points raised during the review process.

**ACADEMIC EDITOR:** Please revise the reviewers' comments and modify the manuscript accordingly. In addition, I would like to highlight the need to include a sample size estimation showing how representative your sample is. Also, I suggest validating putative protein markers of oesophageal carcinoma. You can validate some markers by running a secondary analysis using external data, demonstrating their labeling by targeted methods such as western blot, or running an independent validation with additional samples.

We look forward to receiving your revised manuscript.

Kind regards,

Alexis G. Murillo Carrasco

Academic Editor

PLOS ONE

Journal Requirements:

 [This study was supported by a Cancer Research UK Centres Network Accelerator Award Grant (C328/A21998). Instrumentation in the Centre for Proteomic Research is supported by the BBSRC (BM/M012387/1).]. 

4. Please expand the acronym “BBSRC” (as indicated in your financial disclosure) so that it states the name of your funders in full.

Reviewers' comments:

Reviewer's Responses to Questions

**Comments to the Author**

1. Is the manuscript technically sound, and do the data support the conclusions?

Reviewer #1: Yes

Reviewer #2: Yes

Reviewer #3: Yes

Reviewer #4: Partly

2. Has the statistical analysis been performed appropriately and rigorously?

Reviewer #1: I Don't Know

Reviewer #2: Yes

Reviewer #3: Yes

Reviewer #4: Yes

3. Have the authors made all data underlying the findings in their manuscript fully available?

Reviewer #1: Yes

Reviewer #2: Yes

Reviewer #3: Yes

Reviewer #4: Yes

4. Is the manuscript presented in an intelligible fashion and written in standard English?

Reviewer #1: Yes

Reviewer #2: No

Reviewer #3: Yes

Reviewer #4: Yes

Reviewer #1: Your PICO isn't very clear in your abstract noe introduction, I would recommend clarifying it. I would recommend splitting your "Results" section to methods, analysis, and results instead of joining it all into one

Reviewer #2: Overall, this is an interesting observation study however the major limitations are first the sample size and second the lack of further analysis. The study would benefit from a more detailed focus in the discussion and avoid usign rather generic statements such as 'suggesting that tumourigenesis may impact cellular phenotype'.

There are also quite a number of typographical errors - a few examples below:

spelling- 'mass spectrometery proteomics'

missing word and punctuation - For enrichment of 'Reactome pathways [13] also found changes in'

capital letters- 'Bligh EG, Dyer WJ. A RAPID METHOD OF TOTAL LIPID EXTRACTION AND PURIFICATION.'

avoid discussing the results in the results section - for eg 'Anterior Gradient 2 (AGR2) , involved protein

folding and secretion was the most DEP for OAC. High AGR2 gene expression is a known

unfavourable prognostic marker in renal and liver cancer [8].'

avoid single sentence paragraphs

eg 'The principal value of the results presented here is the quantification of 3,500 proteins of which

nearly 500 were differently expressed as a resource to other OAC researchers.'

It is not clear if this refers to a previous study as the references do not correspond, or whether it was done during this study? 'Additionally, the finding that the G95R mutation decreases the stability of NOP58 suggests...'

Reviewer #3: In the manuscript entitled "Comparative analysis of protein expression between oesophageal adenocarcinoma and normal adjacent tissue," the authors present the proteomics analysis of differential protein expression data obtained from mass spectrometry from 7 cancer patients.

The differential protein expression highlights many previously known biomarkers along with a handful of OAC-specific potential candidate proteins across multiple functional classes. Although this kind of analysis will be even more powerful and informative with a larger sample size, I think this study still provides firsthand information about potential candidates for further scrutiny.

Minor queries

The PCA shows that group PC1 contributes significantly higher variation. What is the nature of this group in component analysis?

The relative heterogeneity between the 7 patents seems quite obvious even in the DEP analysis. Does this heterogeneity stem from the small sample size or the personalized nature of each patient?

Reviewer #4: The manuscript "Comparative analysis of protein expression between oesophageal adenocarcinoma and normal adjacent tissue" has been provided by the authors in a comprehensive manner. . The results complement their previous proteogenomics study to identify neoantigens oesophageal adenocarcinoma patients. In this study, differential expression of oesophageal adenocarcinoma and normal adjacent tissue has been carried out. Below are some concerns

1) How did you identify G95R mutation decreases the stability of NOP58?

2) In this study, Is NOP58 differentially expressed that it gives a defective ribosomal product (DRiP)?

3) Apart from G2/M checkpoint related protein, CDC5L others genes/proteins are also differentially expressed, Why only these genes are targeted?

**Do you want your identity to be public for this peer review?** For information about this choice, including consent withdrawal, please see our Privacy Policy

Reviewer #1: No

Reviewer #2: No

Reviewer #3: **Yes: ** Hariharan Parameswaran

Reviewer #4: **Yes: ** Dicky John Davis G

---

## [Author Response · Author response to Decision Letter 1]

18 Dec 2024

We have submitted a Word document with the editor and reviewer comments and author responses.

---

## [Decision Letter · Decision Letter 1]

31 Dec 2024

Dear Dr. Bailey,

We look forward to receiving your revised manuscript.

Kind regards,

Alexis G. Murillo Carrasco

Academic Editor

PLOS ONE

Journal Requirements:

Reviewers' comments:

Reviewer's Responses to Questions

**Comments to the Author**

Reviewer #2: All comments have been addressed

Reviewer #3: All comments have been addressed

2. Is the manuscript technically sound, and do the data support the conclusions?

Reviewer #2: Yes

Reviewer #3: Yes

3. Has the statistical analysis been performed appropriately and rigorously?

Reviewer #2: I Don't Know

Reviewer #3: Yes

4. Have the authors made all data underlying the findings in their manuscript fully available?

Reviewer #2: Yes

Reviewer #3: Yes

5. Is the manuscript presented in an intelligible fashion and written in standard English?

Reviewer #2: Yes

Reviewer #3: Yes

Reviewer #2: There are still some minor corrections that need to be made, for example:

- Notable in NAT were high expression in NAT of

- be consistent when referring to companies and provide countries/headquarters for eg there is Waters, UK but no country etc for other companies.

- (Thermo Fisher Scientific,UK) but other times it is just Thermo Scientific

- use lowercase n for 'specifically Neutrophil degranulation'

I think there should be a bit more emphasis on studies that have previously identified the same proteins eg for AGR in Barretts https://pmc.ncbi.nlm.nih.gov/articles/PMC2575112/

Reviewer #3: The revised draft entitled, "Comparative analysis of protein expression between oesophageal adenocarcinoma and normal adjacent tissue.", addresses my queries raised in the initial submission.

**Do you want your identity to be public for this peer review?** For information about this choice, including consent withdrawal, please see our Privacy Policy

Reviewer #2: No

Reviewer #3: **Yes: ** Hariharan Parameswaran

---

## [Author Response · Author response to Decision Letter 2]

14 Jan 2025

Our responses are in the uploaded response to reviewers document.

---

## [Editor Report · Decision Letter 2]

20 Jan 2025

Comparative analysis of protein expression between oesophageal adenocarcinoma and normal adjacent tissue

PONE-D-24-45100R2

Dear Dr. Bailey,

We’re pleased to inform you that your manuscript has been judged scientifically suitable for publication and will be formally accepted for publication once it meets all outstanding technical requirements.

Kind regards,

Alexis G. Murillo Carrasco

Academic Editor

PLOS ONE
---

## [Editor Report · Acceptance letter]

PONE-D-24-45100R2

PLOS ONE

Dear Dr. Bailey,

I'm pleased to inform you that your manuscript has been deemed suitable for publication in PLOS ONE. Congratulations! Your manuscript is now being handed over to our production team.

Kind regards,

on behalf of

Dr. Alexis G. Murillo Carrasco

Academic Editor

PLOS ONE